# SAM-R1: Leveraging SAM for Reward Feedback in Multimodal Segmentation via Reinforcement Learning

**Jiaqi Huang**[*], **Zunnan Xu**[*], **Jun Zhou**[†], **Ting Liu**, **Yicheng Xiao**,
**Mingwen Ou**, **Bowen Ji**, **Xiu Li**, **Kehong Yuan**
Tsinghua University

## Abstract

Leveraging multimodal large models for image segmentation has become a prominent research direction. However, existing approaches typically rely heavily on manually annotated datasets that include explicit reasoning processes, which are costly and time-consuming to produce. Recent advances suggest that reinforcement learning (RL) can endow large models with reasoning capabilities without requiring such reasoning-annotated data. In this paper, we propose SAM-R1, a novel framework that enables multimodal large models to perform fine-grained reasoning in image understanding tasks. Our approach is the first to incorporate fine-grained segmentation settings during the training of multimodal reasoning models. By integrating task-specific, fine-grained rewards with a tailored optimization objective, we further enhance the model's reasoning and segmentation alignment. We also leverage the Segment Anything Model (SAM) as a strong and flexible reward provider to guide the learning process. With only 3k training samples, SAM-R1 achieves strong performance across multiple benchmarks, demonstrating the effectiveness of reinforcement learning in equipping multimodal models with segmentation-oriented reasoning capabilities.

## 1 Introduction

Multimodal Large Language Models (MLLMs) [17, 20, 37, 22, 5, 44, 51] have achieved remarkable progress in the field of visual understanding [36, 42, 27, 43, 12], with their capabilities extending to more complex and fine-grained perception tasks like multimodal segmentation [38, 21, 40]. Compared to conventional segmentation methods that rely on simple categorical labels, the reasoning segmentation task [2, 16, 31] has garnered significant attention for its flexibility and practical applicability, but it also introduces substantially greater challenges. Specifically, it requires models not only to comprehend the intent behind user-provided textual queries accurately but also to perform strong logical reasoning to generate high-quality, pixel-level segmentation outputs.

LISA [16] was the first to introduce the integration of MLLMs with segmentation models via specialized tokens, demonstrating the feasibility of applying MLLMs to reasoning segmentation tasks. Building on this foundation, subsequent studies [25, 2, 31, 41] have adopted similar strategies, leveraging task-specific tokens generated by MLLMs to improve pixel-level segmentation performance. While these approaches are promising, they often rely heavily on large-scale annotated datasets to jointly fine-tune the language model and the segmentation decoder. This not only increases training costs but also raises the risk of catastrophic forgetting, where models perform well on in-domain data but fail to generalize to out-of-domain scenarios [7]. Furthermore, the reasoning segmentation tasks frequently involve ambiguous and complex text queries from users, which demand strong reasoning

---

[*]Equal contribution
[†]Corresponding author

39th Conference on Neural Information Processing Systems (NeurIPS 2025).

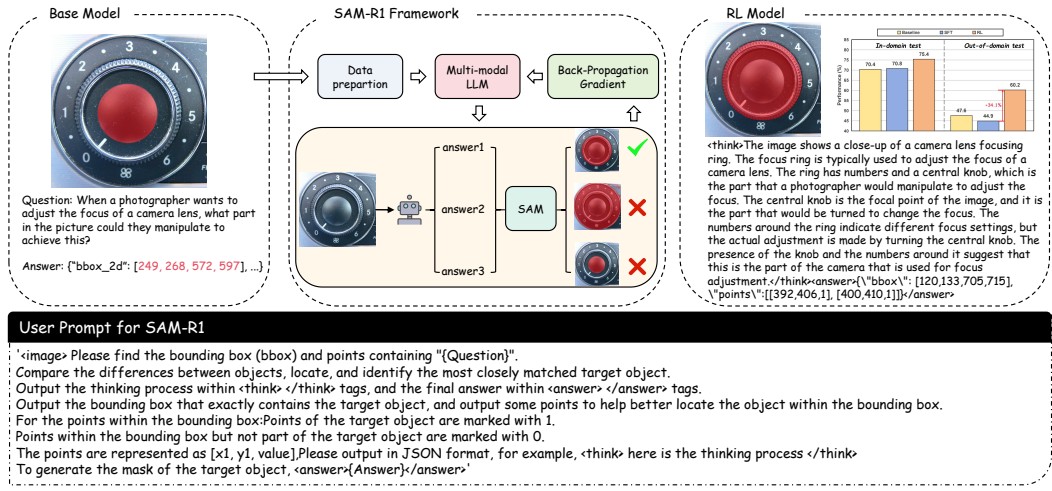

Figure 1: SAM-R1 generates a reasoning chain prior to producing the segmentation mask. It employs a reinforcement learning (RL) strategy, learning the reasoning process from scratch. In comparison to supervised fine-tuning (SFT), the RL-enhanced model, which incorporates fine-grained rewards based on SAM, demonstrates superior performance on both in-domain and out-of-domain data.

capabilities from MLLMs to accurately interpret intent and precisely localize the target segmentation regions.

Recent research has shown that reinforcement learning (RL) can significantly enhance the reasoning capabilities of large language models (LLMs) through reward-based feedback mechanisms [13]. DeepSeek-R1 [9] leverages rule-based rewards to further improve the model's capacity for complex reasoning. This method requires the model to undergo an extensive reasoning process before producing a final answer, with rewards assigned solely based on the correctness of the final response and its adherence to a predefined output format. Such rule-based reward designs align naturally with visual understanding tasks, which often come with accurate ground-truth (GT) annotations. Inspired by this, numerous efforts [8, 46, 50, 39] have applied Group Relative Policy Optimization (GRPO) [33] to vision-language models, incorporating task-specific reward signals. For example, VLM-R1 [34] introduces both format and accuracy rewards for general vision-language tasks, and further incorporates customized rewards tailored to specific applications to mitigate reward hacking. Seg-Zero [23] expands this paradigm by designing a more comprehensive reward system, including reasoning-format, segmentation-format, and accuracy rewards based on IoU and L1 distance, to stimulate robust reasoning in segmentation contexts. Although Seg-Zero demonstrates strong performance in emergent reasoning tasks, its complete decoupling of the reasoning model and segmentation decoder prevents access to pixel-level feedback from the segmentation results, thereby increasing the risk of reward hacking. To address this, involving the segmentation decoder directly in the reward loop, as a reward provider, not only ensures alignment between optimization objectives and task goals but also alleviates the need for extensive human-annotated reasoning data, enabling a more efficient and scalable learning paradigm.

Building upon the insights presented above, we propose SAM-R1, an efficient end-to-end framework tailored for reasoning segmentation. SAM-R1 utilizes reinforcement learning with reward-driven optimization to enhance the reasoning capabilities of MLLMs in complex scenarios. A key component of our framework is the design of task-specific, fine-grained reward functions, particularly a segmentation accuracy reward derived directly from the output of the Segment Anything Model (SAM). This enables the model to develop fine-grained perceptual reasoning in an end-to-end manner—an aspect that has been largely overlooked in previous multimodal reasoning models for segmentation. Integrating powerful SAM [15] has become a prevalent strategy for achieving precise pixel-level segmentation. SAM's zero-shot segmentation capabilities, facilitated by flexible prompt-based inputs, render it a highly adaptable component. While existing approaches often employ SAM as a downstream module to generate segmentation masks based MLLM outputs [23], our framework distinguishes itself by incorporating SAM directly into the reinforcement learning training loop as a reward signal generator. This integration enables the MLLM to receive direct, task-relevant feedback

based on segmentation accuracy, thereby aligning model optimization with the final task objective in a principled and effective manner.

Moreover, we introduce a subtle modification to the clipped objective of PPO to fully utilize its potential in the reasoning segmentation task. First, we increase the upper clipping threshold to encourage updates from highly advantageous actions, thereby granting the model greater flexibility in optimizing the task-specific reasoning model. Second, we observe that GRPO may occasionally produce overly lengthy responses with limited informative content. During the GRPO optimization process, overly long responses will confuse the model and prevent it from obtaining higher reward signals, which can lead to reward hacking. Rather than constraining each token in a single response, we treat all tokens within a response group, encouraging the policy model to focus on generating responses with higher information density. By integrating task-specific, fine-grained rewards with a tailored optimization objective, SAM-R1 precisely interprets complex instructions and accurately localizes segmentation targets. Using only 3K training samples, our method surpasses the base model by 34.1% on the challenging ReasonSeg benchmark in zero-shot setting. In conclusion, our contributions can be summarized as below:

- We present a novel end-to-end framework for fine-grained, reasoning segmentation that employs rule-based rewards to enhance comprehension of complex instructions.

- We devise task-specific, fine-grained reward functions that leverage SAM as an active reward provider, driving continuous self-improvement of the reasoning model.

- We provide extensive empirical evidence demonstrating the effectiveness of SAM-R1 and offer new insights into synergizing reinforcement learning with MLLMs.

## 2 Related Works

### 2.1 MLLMs for Vision and Reasoning Segmentation

Multimodal Large Language Models (MLLMs) have significantly advanced visual understanding, extending from foundational tasks like image captioning and visual question answering [1, 26, 6] to more intricate, fine-grained perception challenges such as image segmentation. A notable direction is reasoning segmentation [45, 2, 31], which necessitates that models interpret implicit user queries and perform logical deduction to generate pixel-level masks. A relevant line of research [10, 35, 11] focuses on using a single generic prompt to perform segmentation, thereby reducing the reliance on manually provided, image-specific inputs. Seminal works like LISA [16] demonstrated the viability of MLLMs for such tasks by interfacing them with segmentation models via specialized tokens. However, these initial approaches frequently depended on Supervised Fine-Tuning (SFT) using datasets with simple categorical labels or rudimentary descriptions [23]. This reliance often curtailed out-of-domain generalization and lacked explicit, interpretable reasoning processes [23, 34], thereby motivating the exploration of methods to instill more robust reasoning capabilities within MLLMs for segmentation.

### 2.2 RL for Enhanced Reasoning in Multimodal Tasks

Reinforcement Learning (RL) has emerged as a potent methodology for eliciting and augmenting the reasoning capacities of large models, circumventing the need for datasets with explicit reasoning annotations. Research indicates that reward-driven optimization can effectively activate emergent test-time reasoning. Algorithms such as Group Relative Policy Optimization (GRPO) [33], employed in models like DeepSeek-R1 for language tasks [9], Seg-Zero for reasoning segmentation [23], and VLM-R1 [34] for general vision-language tasks, have achieved considerable success in training models to generate reasoning chains and attain high performance with limited supervision. These RL-based strategies often exhibit superior generalization compared to SFT methods [7], which are prone to overfitting and catastrophic forgetting of general abilities. Our work leverages this paradigm by adapting an RL training algorithm based on GRPO [33], specifically tailored to the multimodal segmentation task, to cultivate fine-grained perceptual reasoning.

## 2.3 Segmentation Feedback with Task-Specific Rewards

The incorporation of powerful, pre-trained segmentation models like the Segment Anything Model (SAM) [15] has become a prevalent strategy for achieving precise pixel-level segmentations. SAM's zero-shot segmentation capabilities, prompted by diverse inputs, render it a versatile component. While many frameworks employ SAM as a downstream module to produce segmentation masks from MLLM outputs [23], our approach uniquely integrates SAM as an active element within the RL training loop, functioning as a reward provider. This allows the MLLM to receive direct feedback on the quality of its generated information, assessed by the final segmentation accuracy.

The design of effective reward functions is paramount in RL. Related works often employ rule-based rewards, encompassing format rewards for structured outputs and accuracy rewards (e.g., Intersection over Union (IoU) for bounding boxes or masks, L1 distance) to quantify the quality of spatial predictions. For instance, Seg-Zero utilizes reasoning-format, segmentation format, and accuracy rewards based on IoU and L1 distance [23]. VLM-R1 also employs accuracy and format rewards for tasks such as referring expression comprehension and open-vocabulary object detection [34]. Other works like RM-R1 focus on correctness-based rewards for reward modeling itself [3], and R1-Reward introduces consistency rewards alongside formatting and result rewards for training multimodal reward models [50]. Our SAM-R1 framework is distinguished by its design of task-specific, fine-grained reward functions, notably a segmentation-accuracy reward that directly utilizes SAM's output. This enables the model to learn fine-grained reasoning for segmentation tasks in an end-to-end manner, an aspect largely overlooked in prior work on fine-grained segmentation settings within multimodal reasoning models.

# 3 Method

In this section, we elaborate on the architecture of our framework. In section 3.1, we explain how our framework enables multimodal large models to achieve fine-grained perceptual reasoning capacities. The enhancements made to the reinforcement learning algorithm, which significantly enhance the model's multimodal reasoning performance, are detailed in section 3.2. Furthermore, in section 3.3, we offer a detailed discussion of our approach to designing the reward function, with SAM integrated as a strong and flexible reward provider.

## 3.1 SAM-R1

As depicted in Figure 2, our framework takes user-supplied questions and images as input. It performs reasoning and analysis to pinpoint the target object by synthesizing information from both modalities. Subsequently, the model generates intermediate reasoning outputs, which serve as inputs to the segmentation model for mask generation. During this process, the model has the flexibility to produce outputs that enhance the segmentation model's performance. Our approach diverges from prior work [23], which centered on training the multimodal large model alone. Instead, we incorporate the segmentation model as a reward provider in the reinforcement learning phase. This integration enables the segmentation model's outputs to offer detailed feedback, thereby refining the training of the reasoning model.

## 3.2 RL Training Algorithm

Using reinforcement learning [9] to train large models and enhance their performance in specific domains, such as mathematics and programming, has proven effective. However, previous reinforcement learning methods often relied on a pre-trained model, which led to a significant increase in cost and complexity. At the same time, acquiring reasoning capabilities previously required carefully curated datasets that included explicit reasoning processes. Models needed to be trained on these reasoning-annotated datasets to achieve competitive performance.

Recent research [9] has shown that large models' reasoning abilities can merge even when trained on datasets without explicit reasoning rules, and the reward mechanism can be greatly simplified while still maintaining the model's strong performance.

### 3.2.1 DeepSeek R1-Zero and GRPO

The DeepSeek R1-Zero algorithm introduces a novel training approach using Group Relative Policy Optimization (GRPO). This method trains the model to output both a reasoning process and a final answer, while supervision is applied only to the answer. Despite this limited supervision, the model still achieves robust reasoning performance. In this framework, rule-based and accuracy-based reward functions are used to evaluate the model's responses, effectively preventing reward hacking and simplifying the overall reward mechanism.

Unlike previous reinforcement learning algorithms such as PPO [32], which require a separate critic model to evaluate performance, GRPO eliminates the need for an additional model by directly comparing all scores within a group as a baseline. Specifically, for each input question $q$, GRPO samples a set of $G$ responses $\{o_1, o_2, \ldots, o_G\}$ from the old policy $\pi_{\theta_{old}}$. The reward advantage $A_i$ for the $i$-th response is then computed by normalizing the group of rewards $\{r_1, r_2, \ldots, r_G\}$:

$$A_i = \frac{r_i - \mu_r}{\sigma_r}, \tag{1}$$

where $\mu_r$ and $\sigma_r$ denote the mean and standard deviation of the rewards in the group, respectively.

Similar to PPO, GRPO adopts a clipped objective, together with a directly imposed KL penalty term:

$$\mathcal{J}_{GRPO}(\theta) = \mathbb{E}[q \sim P(Q), \{o_i\}_{i=1}^G \sim \pi_{\theta_{old}}(O|q)]$$

$$\left[ \frac{1}{G} \sum_{i=1}^G \frac{1}{|o_i|} \sum_{t=1}^{|o_i|} \left( \min\left( \frac{\pi_\theta(o_i|q)}{\pi_{\theta_{old}}(o_i|q)} A_i, \text{clip}\left( \frac{\pi_\theta(o_i|q)}{\pi_{\theta_{old}}(o_i|q)}, 1-\varepsilon, 1+\varepsilon \right) A_i \right) - \beta \mathbb{D}_{KL}\left( \pi_\theta || \pi_{ref} \right) \right) \right], \tag{2}$$

where the KL divergence is defined as:

$$\text{D}_{KL}\left( \pi_\theta || \pi_{ref} \right) = \frac{\pi_{ref}(o_i|q)}{\pi_\theta(o_i|q)} - \log \frac{\pi_{ref}(o_i|q)}{\pi_\theta(o_i|q)} - 1. \tag{3}$$

### 3.2.2 Our Training Algorithm

Similar to recent studies [49, 24], we observe that the clipping term utilized in advantage estimation is beneficial for maintaining stability in policy updates. At the same time, the KL-divergence penalty already limits the distributional shift between successive policies and therefore also serves as a stabilizing factor. In our multimodal image-segmentation task, we aim to allow the large multimodal model greater freedom to explore finer-grained interpretations while preserving training stability. Hence, we retain the KL constraint but decouple the clipping mechanism: we replace the single

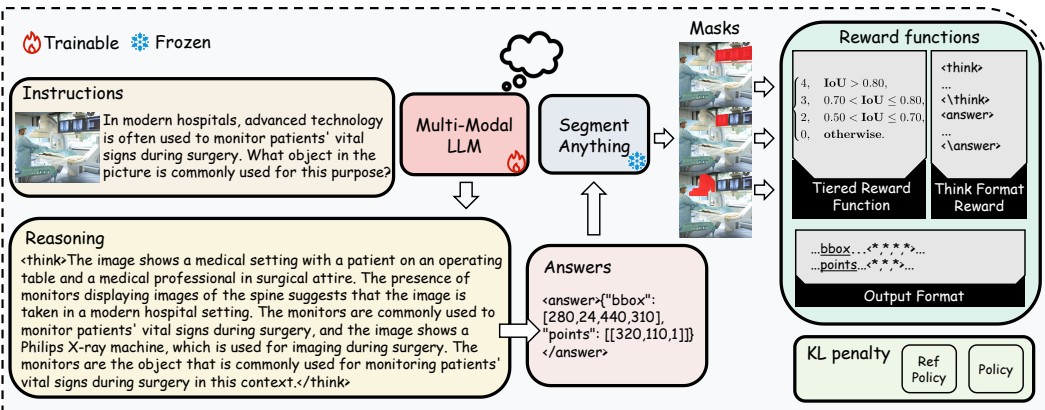

Figure 2: Our framework integrates the Segment Anything Model (SAM) as a reward provider in the reinforcement learning training of a multimodal large model (MLLM). The two models jointly process user-input questions and images to identify target objects and generate masks. Specifically, the MLLM generates the reasoning process and answer, then passes them to SAM. A fine-grained reward based on Intersection over Union (IoU) is calculated to optimize the MLLM.

threshold $\varepsilon$ with asymmetric bounds $\varepsilon_{\text{low}}$ and $\varepsilon_{\text{high}}$. We keep $\varepsilon_{\text{low}}$ unchanged and slightly raise $\varepsilon_{\text{high}}$ to encourage broader exploration.

We also observe that GRPO can sometimes yield very long yet low-information answers. Such responses waste tokens and increase the risk of hallucination, as long and short answers incur the same total loss, thereby causing the per-token penalty for longer responses. To counter this, we rescale the loss so that every token receives the same loss, discouraging redundant and repetitive outputs. With these changes, our training objective becomes:

$$
\mathcal{J}_{ours}(\theta) = \mathbb{E}[q \sim P(Q), \{o_i\}_{i=1}^{G} \sim \pi_{\theta_{\text{old}}}(O|q)]
$$
$$
\left[ \frac{1}{\sum_{i=1}^{G}|o_i|} \sum_{i=1}^{G} \sum_{t=1}^{|o_i|} \left( \min \left( \frac{\pi_\theta(o_i|q)}{\pi_{\theta_{\text{old}}}(o_i|q)} A_i, \right. \right. \right.
$$
$$
\left. \left. \left. \text{clip} \left( \frac{\pi_\theta(o_i|q)}{\pi_{\theta_{\text{old}}}(o_i|q)}, 1 - \varepsilon_{\text{low}}, 1 + \varepsilon_{\text{high}} \right) A_i \right) - \beta \mathbb{D}_{KL} \left( \pi_\theta \| \pi_{\text{ref}} \right) \right) \right]. \tag{4}
$$

These modifications allow the model to explore aggressively, achieve a fine-grained understanding, and train stably without incurring the extra cost and complexity of an additional critic model.

## 3.3  Reward Functions

A reward model is a crucial component of reinforcement learning (RL): combined with preference-alignment algorithms, it steers the policy toward the desired objectives. Following earlier work [9], we likewise employ reward functions and adapt them to the multimodal segmentation setting through three task-specific, rule-based rewards.

**Tiered Segmentation-accuracy Reward Function.** Departing from earlier reward designs, we treat SAM (Segment Anything Model) as an external reward provider. The target location predicted by the multimodal model is passed to SAM, which returns a mask prediction. We compute the IoU between this mask and the ground-truth mask and assign piecewise rewards:

$$
\text{reward} = \begin{cases} 4, & \text{IoU} > 0.80, \\ 3, & 0.70 < \text{IoU} \leq 0.80, \\ 2, & 0.50 < \text{IoU} \leq 0.70, \\ 0, & \text{otherwise}, \end{cases} \tag{5}
$$

which provides robust positive feedback only when the predicted region closely aligns with the ground truth, guiding the model toward gradual improvement at lower IoU levels.

**Reasoning-format reward.** To encourage explicit reasoning, the model should enclose its chain-of-thought between "<think>" and "</think>" tags and place the final answer between "<answer>" and "</answer>" tags. Outputs that adhere to this structure receive a positive reward, while malformed outputs incur a penalty.

**Segmentation-format reward.** To ensure the multimodal large model provides fine-grained cues to the downstream segmentation module, it must emit the detected bounding box, a reference point, and a descriptive textual flag in a prescribed JSON-like format. Compliance with the schema yields a reward; deviations incur a penalty.

## 4  Experiment

### 4.1  Experimental Setup

We use Qwen2.5VL-7B [1] as our base model and SAM2-Large [29] as the segmentation model. All experiments are conducted on $8 \times$A100 GPUs. During training, we sample 8 responses per question, set $\varepsilon_{\text{high}} = 0.3$, and use a learning rate of $1.0 \times 10^{-6}$. To ensure the model's robustness across different domains, we resize all images to $840 \times 840$ before feeding them into the MLLM during both training and evaluation. We follow previous works [16, 23] and use both cIoU and gIoU as evaluation metrics. gIoU is defined by the average of all per-image intersection-over-unions, while cIoU is defined by the cumulative intersection over the cumulative union.

## 4.2 Datasets

For training, we randomly sample 3,000 instances from the training set of RefCOCOg [48], which contains 104,560 referring expressions tied to 54,822 objects across 26,711 images. We use the official RefCOCOg test set as our in-domain evaluation set. To assess generalization across datasets, we use the testA subsets from RefCOCO and RefCOCO+ [14] as our out-of-distribution (OOD) evaluation sets. RefCOCO consists of 142,210 expressions for 50,000 objects across 19,994 images, while RefCOCO+ includes 141,564 expressions for 49,856 objects in 19,992 images, with both datasets providing predefined splits. RefCOCO+ is considered more challenging due to the exclusion of absolute location terms. In addition, we include ReasonSeg [16], a dataset that requires strong visual-linguistic reasoning, to further evaluate our model's ability to perform fine-grained segmentation under complex reasoning conditions.

## 4.3 Main Results

**ReasonSeg.** Table 1 shows the zero-shot performance of SAM-R1 on the ReasonSeg benchmark. Our method achieves 60.2% gIoU and 54.3% cIoU on the test set, outperforming the previous best, Seg-Zero (58.3% gIoU and 53.4% cIoU). This improvement is mainly due to our fine-grained reward design, which integrates SAM into the RL loop to provide IoU-based feedback during training, aligning reasoning with segmentation. Unlike Seg-Zero's decoupled design, our unified framework introduces finer-grained segmentation rewards, enabling stable optimization and better generalization with only 3k training samples. Additionally, our improved GRPO strategy—with asymmetric clipping and token-level loss normalization—enhances informativeness and robustness under domain shifts, supporting SAM-R1's strong zero-shot performance in complex reasoning segmentation. Seg-Zero-7B* denotes performance based on provided model weights, as their reported results used different weights per metric and could not be reproduced.

Table 1: Comparison on ReasonSeg-zero-shot benchmark (val/test). The best results are in bold.

| Method | ReasonSeg-zero-shot | | | |
| | val | | test | |
| | gIoU | cIoU | gIoU | cIoU |
| --- | --- | --- | --- | --- |
| OVSeg [18] | 28.5 | 18.6 | 26.1 | 20.8 |
| ReLA [19] | 22.4 | 19.9 | 21.3 | 22.0 |
| Grounded-SAM [30] | 26.0 | 14.5 | 21.3 | 16.4 |
| LISA-7B-LLaVA1.5 [16] | 53.6 | 52.3 | 48.7 | 48.8 |
| LISA-13B-LLaVA1.5 [16] | 57.7 | 60.3 | 53.8 | 50.8 |
| SAM4MLLM [4] | 46.7 | 48.1 | - | - |
| Seg-Zero-7B* [23] | 62.0 | 52.0 | 58.3 | 53.4 |
| **SAM-R1 (Ours)** | **64.0** | **55.8** | **60.2** | **54.3** |

**Referring Expression Segmentation.** Our evaluation results on the Referring Expression Segmentation datasets are shown in Table 2. We use the testA subsets of RefCOCO and RefCOCO+ as OOD test sets, and the test set of RefCOCOg as the in-domain test set. It can be seen that our model, trained on only 3,000 samples, still achieves competitive performance compared to prior methods. Specifically, on the in-domain dataset RefCOCOg, our algorithm SAM-R1 is only 0.2 points lower than Seg-Zero, despite using fewer style-consistent training samples. On the OOD datasets, our model performs comparably to Seg-Zero on RefCOCO, and improves the performance on RefCOCO+ from 73.9 to 74.7. This demonstrates the effectiveness of our approach SAM-R1. We attribute this improvement to the fine-grained reward mechanisms and the flexible exploration strategy, which allows our model to surpass previous out-of-domain performance with significantly less training data.

## 4.4 Visualization Analysis

As shown in Figure 3, we present some representative cases to analyze the reasoning and segmentation performance of our model in diverse scenarios.

**Multiple Subjects with Fine-Grained Segmentation.** In certain situations, it is necessary to identify a specific subject among multiple subjects. For example, identifying Santa Claus amidst a little girl,

Table 2: Performance comparison on referring expression benchmarks using cIoU.

| Method | refCOCO | refCOCO+ | refCOCOg |
|---|---|---|---|
| LAVT [47] | 75.8 | 68.4 | 62.1 |
| ReLA [19] | 76.5 | 71.0 | 66.0 |
| LISA-7B [16] | 76.5 | 67.4 | 68.5 |
| PixelLM-7B [31] | 76.5 | 71.7 | 70.5 |
| PerceptionGPT-7B [28] | 78.6 | 73.9 | 71.7 |
| Seg-Zero-7B* [23] | **79.2** | 73.9 | **73.3** |
| **SAM-R1 (Ours)** | **79.2** | **74.7** | 73.1 |

chairs, Christmas trees, and various decorations, each of which is complex and numerous. The model utilizes cues, such as red clothing and the act of listening to wishes, to successfully identify and segment Santa Claus.

**Global To Local Reasoning.** In scenes containing rich local details, identifying a specific part from the overall structure is highly challenging. For example, in an image of an airplane composed of various components, our model accurately locates the engine by reasoning over the spatial relationship between the engine and the wings.

**Challenging Environment With Distractors.** In cluttered environments, such as an airport filled with various signs, identifying a specific sign, such as "Watch Your Step", poses significant challenges. Our model effectively distinguishes the target sign from visually similar ones by leveraging contextual reference objects and localizing the identification process step by step.

**Complex Boundaries.** For complex boundaries, such as those found in gymnastics competitions, the model integrates textual and visual information to infer that gymnastics involves specific movements. This understanding suggests the use of a vaulting table, which in turn facilitates the generation of coherent segmentation masks.

### 4.5 Ablation Study

In this section, we validate the effectiveness of the proposed components. As shown in Table 3, the tiered threshold strategy demonstrates superior performance compared to fixed thresholds across both in-domain and OOD benchmarks. While fixed thresholds of 0.5, 0.7, and 0.8 achieve 56.5-58.6 gIoU on the ReasonSeg-zero-shot (test), the dynamic tiered approach significantly outperforms them with 60.2 gIoU (+3.5% absolute improvement). This performance gap highlights the limitations of static thresholds in handling complex reasoning scenarios, where overly conservative predictions at high thresholds (e.g., 0.8) degrade cIoU performance despite improved localization precision. The tiered mechanism's phased threshold adjustment seems to more effectively balance precision-recall trade-offs, particularly enhancing OOD generalization, as demonstrated by its 75.4 gIoU score on the refCOCOg test set, which is 0.8% higher than the best fixed threshold.

Table 3: Ablation study on different threshold strategies under ReasonSeg-zero-shot (test) and refCOCOg-test benchmarks.

| Method | ReasonSeg (gIoU) | ReasonSeg (cIoU) | refCOCOg (gIoU) | refCOCOg (cIoU) |
|---|---|---|---|---|
| 0.5 | 56.5 | 51.9 | 74.7 | 72.8 |
| 0.7 | 56.2 | 51.6 | 74.9 | 72.6 |
| 0.8 | 58.6 | 50.8 | 74.6 | 71.9 |
| Tiered | 60.2 | 54.3 | 75.4 | 73.1 |

We further analyze the algorithmic components presented in Table 4 to validate the effectiveness of token-level constraints and the use of an asymmetric clipping strategy. The token-level reward mechanism yields consistent improvements across various metrics, enhancing ReasonSeg cIoU by 0.5% (from 51.2% to 51.7%) and refCOCOg cIoU by 0.6% (from 71.8% to 72.4%) through fine-grained output format regulation. Meanwhile, increasing the upper clipping threshold in our GRPO

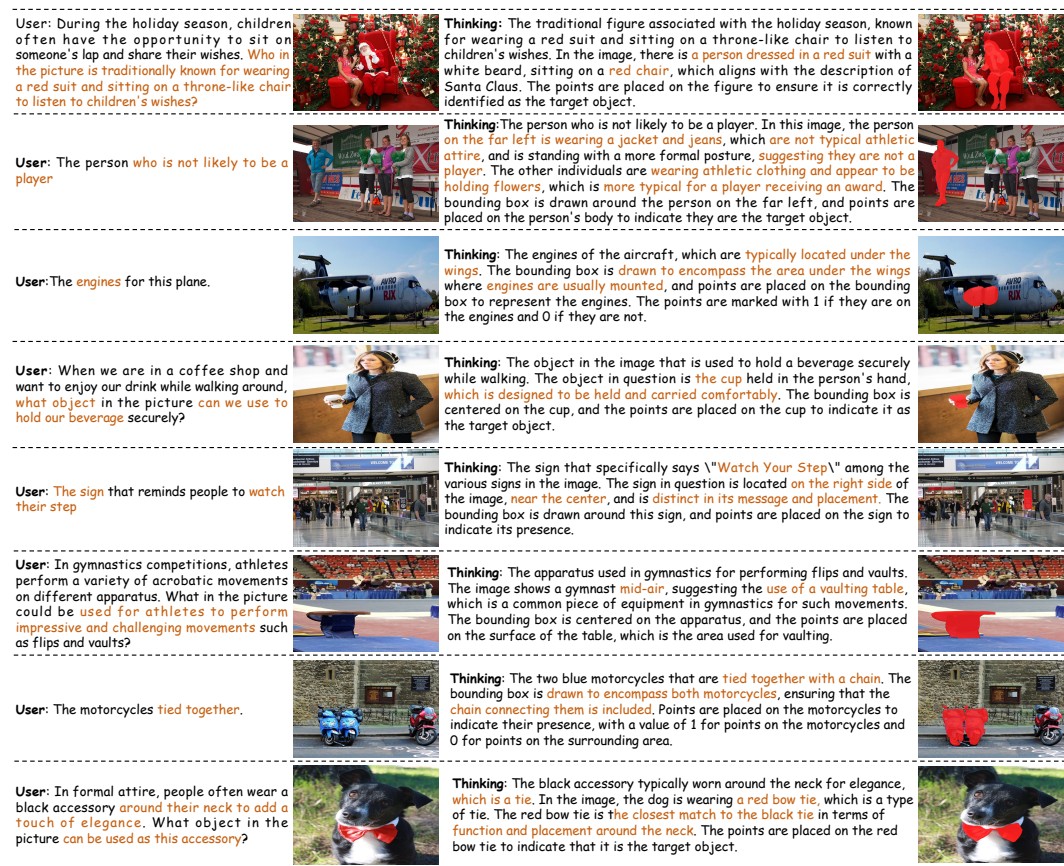

Figure 3: Qualitative results on ReasonSeg [16] demonstrate that SAM-R1 exhibits robust zero-shot performance, further enhanced by the chain-of-thought approach with improved reasoning capacity.

variant provides more flexibility in updating highly advantageous actions, which proves especially beneficial in OOD reasoning tasks. This adjustment improves ReasonSeg gIoU by 1.3%, compared to a 0.8% gain on refCOCOg, suggesting that such flexibility is more impactful in addressing complex reasoning challenges. Notably, combining both techniques yields a synergistic effect, raising ReasonSeg cIoU to 54.3%, a 3.1% improvement over the GRPO baseline. The full method achieves peak gIoU scores of 75.4 on the refCOCOg test set and 60.2 on ReasonSeg, demonstrating the effectiveness of jointly enforcing fine-grained output structure and reward-sensitive policy adaptation.

Table 4: Ablation study of algorithmic components based on the GRPO baseline on ReasonSeg-zero-shot and refCOCOg-test.

| Method | Token-level | CLIP higher | gIoU (RS) | cIoU (RS) | gIoU (Rcg) | cIoU (Rcg) |
|--------|-------------|-------------|-----------|-----------|------------|------------|
| GRPO | ✘ | ✘ | 57.8 | 51.2 | 74.1 | 71.8 |
| Token-level | ✔ | ✘ | 58.0 | 51.7 | 74.5 | 72.4 |
| Clip higher | ✘ | ✔ | 59.1 | 52.8 | 74.9 | 72.5 |
| Ours | ✔ | ✔ | 60.2 | 54.3 | 75.4 | 73.1 |

## 4.6 Generalization to REC task

Although our model is not trained on any Referring Expression Comprehension (REC) datasets, we observe strong performance on REC task, thanks to the model's enhanced reasoning ability and fine-grained perceptual capabilities. As shown in Table 5, our method, SAM-R1, achieves state-of-the-art performance on the LISA-Grounding benchmark with 63.8, significantly surpassing previous methods such as GroundedSAM (26.2), OV-Seg (28.4), X-Decoder (28.5), and Visual-RFT (43.9).

| Model | LISA-Grounding |
|---|---|
| GroundedSAM | 26.2 |
| OV-Seg | 28.4 |
| X-Decoder | 28.5 |
| Visual-RFT | 43.9 |
| SAM-R1(Ours) | **63.8** |

Table 5: Performance comparison on the LISA-Grounding benchmark. Our method significantly outperforms prior open-vocabulary and vision-language segmentation approaches, demonstrating strong generalization ability on reasoning-intensive REC tasks.

This substantial improvement demonstrates the effectiveness of our reinforcement learning-based reasoning framework in complex visual grounding tasks. Unlike prior approaches, which often rely on large-scale supervised training or handcrafted prompt engineering, our method leverages task-aligned rewards and structured reasoning supervision to enable fine-grained object understanding and robust generalization in reasoning-intensive scenarios. These results demonstrate the generality and adaptability of our method beyond segmentation, highlighting its strong alignment capabilities and transferability to challenging REC scenarios.

### 4.7 Broader Impact and Discussion

Our work shows that reinforcement learning, guided by a segmentation model, can effectively cultivate reasoning in multimodal models. The strong performance of SAM-R1 with only 3,000 training samples highlights a promising path toward data efficiency. By using standard segmentation masks as the supervisory signal, our approach bypasses the need for costly and potentially biased, manually annotated reasoning chains, thus enhancing scalability. More broadly, this study supports a paradigm where models learn complex reasoning from task-aligned rewards rather than explicit instructions. This shift toward learning from weaker, accessible supervision is particularly impactful for domains with scarce reasoning data, such as robotic perception and medical image analysis.

We recognize several limitations for future work. First, SAM's parameters remain frozen, creating a one-way information flow that prevents it from adapting to the reasoning model. Jointly optimizing both models is a compelling next step. Though computationally demanding, this could foster a synergistic alignment where the models co-adapt. Second, our model struggles to generate meaningful negative reference points, a key capability for robust discriminative reasoning. Our RL framework failed to encourage this, suggesting a foundational limitation that may require new architectural or algorithmic solutions to improve robustness in complex visual scenes.

## 5 Conclusion

In this paper, we present SAM-R1, an innovative framework that leverages reinforcement learning to enhance the reasoning capabilities of multimodal large models for image segmentation. Our method introduces fine-grained segmentation settings into the training process, enabling more precise and task-relevant reasoning. Furthermore, we propose a task-specific, fine-grained reward design that incorporates the Segment Anything Model (SAM) as a flexible and reliable reward provider. By integrating these components with a tailored optimization objective, SAM-R1 achieves strong performance using only 3,000 training samples, demonstrating the practicality and effectiveness of reinforcement learning in this domain. This work not only contributes to advancing multimodal image segmentation but also highlights the potential of reward-guided learning for developing more efficient and adaptable multimodal large models.

## 6 Acknowledgements

This work was supported by the STI 2030-Major Projects under Grant 2021ZD0201404.

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

# A    Technical Appendices

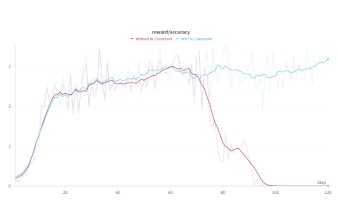
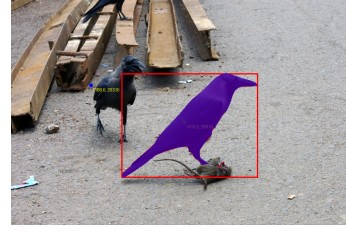
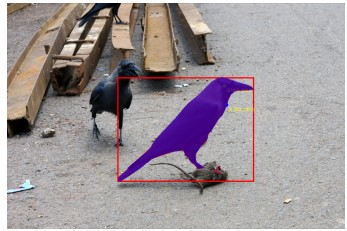

(a) Training collapse without KL constraint.

(b) Negative points placed outside bounding boxes.

(c) All points clustered on the object when restricted to box.

Figure 4: Ablation study failures: (a) Removing the KL constraint leads to training instability and collapse. (b) Encouraging both positive and negative point generation causes negatives to appear outside target areas. (c) Forcing all points into the bounding box eliminates useful contrast, reducing performance.

## A.1    Ablation Failure: Removing the KL Constraint

During the development of our method, we explored various strategies to encourage broader exploration by the model. One such attempt involved removing the KL divergence constraint, which is commonly used to regularize policy updates and limit deviation from the reference distribution.

However, empirical results showed that eliminating the KL term led to significant instability during training. As illustrated in Figure 4a, the model initially exhibited effective learning behavior with a strong exploratory signal. Yet, after approximately 100 training steps, we observed sharp fluctuations in performance, eventually leading to complete collapse of the training process.

This outcome indicates that the KL constraint plays a crucial role in maintaining training stability, especially in our multimodal reasoning setting. Consequently, we decided to retain the KL divergence term in our final framework, despite its potential to limit aggressive exploration.

## A.2    Ablation Failure: Encouraging Negative Reference Points

In designing the reward function, we initially allowed the multimodal large model to freely determine the value of the reference point—positive (1) or negative (0)—without explicit supervision. However, we observed that the model strongly preferred generating only positive points, rarely including any negatives. We hypothesized that incorporating both positive and negative points could provide richer target information and improve segmentation performance.

To encourage this behavior, we introduced a format-based reward component, point value, which awarded 1 point when both 0 and 1 values appeared in the output. As shown in Figure 4b, this led the model to include both types of points. While the positive points remained well-aligned with the target object, the negative points were typically placed at the image boundaries, far outside the bounding box, offering no useful contrast for object discrimination.

We then modified the rule to grant the reward only when both positive and negative points were located within the bounding box. As shown in Figure 4c, this adjustment led to all points—regardless of label—being clustered directly on the target object, effectively eliminating the intended contrast and introducing noise instead.

These results suggest that, despite reward incentives, the multimodal large model lacks the inherent ability to identify meaningful negative examples in visual space. Therefore, we decided not to enforce negative point generation in our final design.

## A.3    Failure Analysis

As illustrated in Figure 5, we present several typical failure cases, which mainly highlight two issues: incomplete segmentation and over-segmentation.

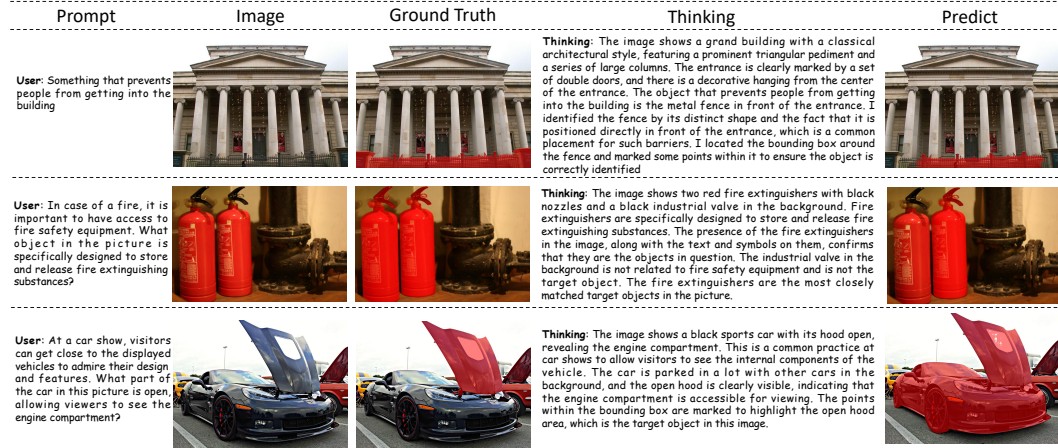

| Prompt | Image | Ground Truth | Thinking | Predict |
|--------|-------|--------------|----------|---------|
| **User**: Something that prevents people from getting into the building | | | **Thinking**: The image shows a grand building with a classical architectural style, featuring a prominent triangular pediment and a series of large columns. The entrance is clearly marked by a set of double doors, and there is a decorative hanging from the center of the entrance. The object that prevents people from getting into the building is the metal fence in front of the entrance. I identified the fence by its distinct shape and the fact that it is positioned directly in front of the entrance, which is a common placement for such barriers. I located the bounding box around the fence and marked some points within it to ensure the object is correctly identified | |
| **User**: In case of a fire, it is important to have access to fire safety equipment. What object in the picture is specifically designed to store and release fire extinguishing substances? | | | **Thinking**: The image shows two red fire extinguishers with black nozzles and a black industrial valve in the background. Fire extinguishers are specifically designed to store and release fire extinguishing substances. The presence of the fire extinguishers in the image, along with the text and symbols on them, confirms that they are the objects in question. The industrial valve in the background is not related to fire safety equipment and is not the target object. The fire extinguishers are the most closely matched target objects in the picture. | |
| **User**: At a car show, visitors can get close to the displayed vehicles to admire their design and features. What part of the car in this picture is open, allowing viewers to see the engine compartment? | | | **Thinking**: The image shows a black sports car with its hood open, revealing the engine compartment. This is a common practice at car shows to allow visitors to see the internal components of the vehicle. The car is parked in a lot with other cars in the background, and the open hood is clearly visible, indicating that the engine compartment is accessible for viewing. The points within the bounding box are marked to highlight the open hood area, which is the target object in this image. | |

Figure 5: Visualization of some failure cases for our SAM-R1 method on the ReasonSeg-val dataset, which shows that our approach still has some limitations.

A notable observation is that our SAM-R1, through its Thinking process (fourth column), successfully comprehends the prompt and correctly localizes the target object(s). However, this correct semantic understanding does not always translate perfectly into the final segmentation mask. For example, in cases of incomplete segmentation: In the first row, the model correctly identifies the fence, but the Predict mask only covers a small portion of the target. In the second row, the model recognizes The fire extinguishers but incorrectly segments only one of the two instances. Conversely, over-segmentation is demonstrated in the third row: The prompt asks for a part of the car (the hood), and the Thinking process also pinpoints the open hood. However, the model incorrectly segments the entire vehicle instead of just the specified part.

These examples indicate that, while our model performs well in high-level semantic reasoning, limitations still exist in its ability to precisely map this understanding to pixel-level masks, particularly concerning fine-grained segmentation and instance completeness. This remains a key area for future improvement.

### A.4  Data Efficiency and Scalability Analysis

To investigate the scalability and data efficiency of SAM-R1, we conducted additional experiments by increasing the size of the training data from 3k to 10k. The results clearly show that our method is highly data-efficient, with performance saturating at just 3k samples.

We present the direct comparison in Table 6. As shown, increasing the data to 10k results in negligible fluctuations in ReasonSeg: the cIoU shifts slightly from 55.8 to 55.5 on the val split and from 54.3 to 53.9 on the test split. Similarly, on the RefCOCO benchmarks, we observe only marginal gains, which strongly indicates that performance has already plateaued.

Table 6: Data efficiency analysis with 3k vs. 10k training samples.

| Method | ReasonSeg | | RefCOCO Benchmarks | | |
|--------|-----------|------|----------|----------|----------|
| | Val | Test | refCOCO | refCOCO+ | refCOCOg |
| SAM-R1 (Ours, 3k) | 55.8 | 54.3 | 79.2 | 74.7 | 73.1 |
| SAM-R1 (Ours, 10k) | 55.5 | 53.9 | 79.9 | 75.3 | 73.5 |
| **Gain** | -0.3 | -0.4 | +0.7 | +0.6 | +0.4 |

From these results, it is evident that our method's core performance saturates at 3k samples. Given the substantial increase in training cost versus the minimal performance returns, we deliberately chose 3k samples as the optimal trade-off point for demonstrating our method's capabilities.

