# OpenReview forum: "SAM-R1: Leveraging SAM for Reward Feedback in Multimodal Segmentation via Reinforcement Learning"
_NeurIPS.cc/2025/Conference — NeurIPS 2025 poster_

### Official Review · Reviewer_KmuG · 2025-06-25

**Clarity:** 3
**Significance:** 3
**Originality:** 2
**Rating:** 4
**Confidence:** 4

**Summary:**

This paper  propose SAM-R1,  a novel framework that enables multimodal large models to perform fine-grained reasoning in image understanding tasks. It incorporates fine-grained segmentation settings during the training of multimodal reasoning models. Extensive experiments demonstrate its effetiveness.

**Questions:**

1.Design and sensitivity of the reward. The goal of SAM-R1 is to enable the segmentation model to leverage GRPO so that it can exhibit reasoning capabilities without requiring an explicit reasoning chain as ground truth. This makes the design of the reward particularly critical. Specifically, the reward in Equation (5) depends heavily on the chosen threshold. Could you explain how this threshold is determined? Why is the reward discretised into multiple levels? Moreover, would small variations in the threshold value lead to significant differences in the results?

2. Clarification of improvements over GRPO. The method presented in this paper appears to directly apply GRPO to the reasoning segmentation task. What are the main technical contributions or modifications compared to standard GRPO? It would be helpful to highlight the key improvements that make the proposed approach better suited for reasoning segmentation.

**Ethical Concerns:**

["NO or VERY MINOR ethics concerns only"]

**Final Justification:**

I have read the authors' rebuttal, I believe my concern have been addressed mostly.

**Limitations:**

Please refers to weakness and questions sections

**Quality:**

2

**Strengths And Weaknesses:**

Strength:

1.	The paper is well-written and easy to follow.

2.	The visualisation is straightforward, and easy to demonstrate the effectiveness of the proposed approach.

3.	The topic the paper addressed is cut-edging and deserve to explore.

Wekaness:

1. Missing related work. The current setting of task-generic, promptable segmentation also aims to infer image-specific answers given a generic prompt and perform corresponding segmentation. It would be beneficial to add a brief discussion of the literature on this setting [1][2][3]:

[1] Hu, J., Lin, J., Gong, S., et al. Relax image-specific prompt requirement in SAM: A single generic prompt for segmenting camouflaged objects. Proceedings of the AAAI Conference on Artificial Intelligence, 2024, 38(11): 12511–12518.

[2] Tang, L., Jiang, P. T., Shen, Z. H., et al. Chain of visual perception: Harnessing multimodal large language models for zero-shot camouflaged object detection. Proceedings of the 32nd ACM International Conference on Multimedia, 2024: 8805–8814.

[3] Hu, J., Lin, J., Yan, J., et al. Leveraging hallucinations to reduce manual prompt dependency in promptable segmentation. The Thirty-eighth Annual Conference on Neural Information Processing Systems, 2024.

2. Insufficient discussion of innovation. The innovation of this work needs to be further clarified. The proposed method appears to directly apply GRPO to the reasoning segmentation task, so the key differences and improvements over standard GRPO must be more explicitly articulated.

---

> ### Author Rebuttal · Authors · 2025-07-31
>
> We sincerely thank the reviewer for the insightful and constructive comments. We have carefully addressed each point as follows:
>
> 1. Missing related work discussion
>
> Thank you for your valuable suggestion. In the revised version, we will include a dedicated discussion on the task-generic, promptable segmentation setting, particularly focusing on methods that aim to infer image-specific answers based on generic prompts and produce corresponding segmentations. Specifically, we will incorporate the three cited papers into this discussion to better contextualize recent advances in this area.
>
>
> 2. Insufficient clarification of innovation
>
>  Leveraging multimodal large models for image segmentation has become a prominent research direction. However, existing approaches typically rely heavily on manually annotated datasets with explicit reasoning chains, which are costly and time-consuming to construct. Some recent methods have explored reinforcement learning for segmentation, such as Seg-Zero, but they often pre-process segmentation tasks into object detection form and fail to leverage task-specific, fine-grained reward signals.
>
>  In contrast, we propose a novel formulation where SAM is directly employed as a reward provider in the reinforcement learning loop, requiring no special pre-processing of the segmentation dataset. This design enables us to use the predicted segmentation masks as supervision, providing direct, pixel-level feedback to guide reasoning. To complement this, we modify the GRPO optimization objective to improve exploration ability and interpretability, allowing the model to follow complex multimodal instructions and accurately localize the segmentation target. These innovations collectively allow SAM-R1 to achieve fine-grained perceptual reasoning with a simplified, end-to-end training pipeline.
>
> 3. Reward threshold design and sensitivity
>
>  Thank you for these detailed questions regarding our reward function.
>
> The core motivation for our tiered reward design is to provide a stable and informative learning curriculum. A single, fixed threshold creates a challenging dilemma in RL-based training:
>
> ●A high threshold makes rewards too sparse, especially in early training. Most predictions receive no reward, hindering efficient learning and exploration.
>
> ●A low threshold leads to quick reward saturation. Once the model can consistently achieve a coarse segmentation, it receives no further incentive for fine-grained refinement.
>
> Our tiered reward strategy (as defined in Equation 5) elegantly resolves this trade-off. By discretizing IoU into multiple levels, it creates a hierarchical learning path, guiding the model progressively from coarse localization (earning a small reward) to precise segmentation (earning the maximum reward).
>
> The effectiveness of this tiered approach is empirically validated by our ablation study, which we include here for your convenience:
>
> | Method | ReasonSeg (gIoU) | ReasonSeg (cIoU) | refCOCOg (gIoU) | refCOCOg (cIoU) |
> | :--- | :---: | :---: | :---: | :---: |
> | Fixed Threshold: 0.5 | 56.5 | 51.9 | 74.7 | 72.8 |
> | Fixed Threshold: 0.7 | 56.2 | 51.6 | 74.9 | 72.6 |
> | Fixed Threshold: 0.8 | 58.6 | 50.8 | 74.6 | 71.9 |
> | **Tiered (Ours)** | **60.2** | **54.3** | **75.4** | **73.1** |
>
> As the results clearly demonstrate, the tiered strategy consistently and significantly outperforms all fixed-threshold alternatives. On the challenging ReasonSeg dataset, it yields a substantial +1.6 to +4.0 point gain in gIoU and a +2.4 to +3.5 point gain in cIoU compared to the single-threshold methods. This provides clear evidence that the hierarchical reward signal is crucial for achieving superior performance.
>
> 4. Clarification of improvements over standard GRPO
>
> While GRPO provides a solid foundation, our method is not a direct application. We introduced significant, task-specific adaptations that are essential for enabling a model to perform complex reasoning for segmentation.
>
> Our main technical contributions can be summarized in three key areas:
>
> ●Task-Specific Reward Engineering: Standard GRPO is designed for general text generation. We engineered a completely new reward function tailored for vision-language grounding. This includes:
>
> A multi-level, IoU-based reward (Eq. 5) to provide a granular learning signal for segmentation accuracy.
>
> A supplementary format-based reward to enforce correct structural output (i.e., reasoning text followed by segmentation coordinates).
>
> ●Core Algorithmic Enhancements: We modified the GRPO optimization objective itself to improve training stability and performance on this unique task. Our key changes are token-level advantage normalization and an asymmetric clipping mechanism (Clip higher). These are not part of the standard GRPO implementation and are vital for balancing the needs of generating fluent reasoning with precise, numerical coordinate outputs.
>
> ●Unified End-to-End Training: Unlike prior works that separate reasoning and segmentation modules, we train our system jointly using RL. This allows the model to learn a direct mapping from visual input and natural language instructions to segmentation masks, using pixel-level feedback to refine its reasoning process.
>
> The necessity and impact of our algorithmic enhancements are empirically validated in our ablation study. We present the key results from Table 4 of our paper below:
>
> | Method | Token-level | CLIP higher | gIoU (RS) | cIoU (RS) | gIoU (Rcg) | cIoU (Rcg) |
> | :--- | :---: | :---: | :---: | :---: | :---: | :---: |
> | GRPO (baseline) | ✘ | ✘ | 57.8 | 51.2 | 74.1 | 71.8 |
> | Token-level | ✔ | ✘ | 58.0 | 51.7 | 74.5 | 72.4 |
> | Clip higher | ✘ | ✔ | 59.1 | 52.8 | 74.9 | 72.5 |
> | **Ours (Full)** | **✔**| **✔** | **60.2** | **54.3** | **75.4** | **73.1** |
>
> As the table clearly shows, the standard GRPO baseline yields the lowest performance. Adding our proposed token-level normalization or asymmetric clipping individually provides noticeable gains. When combined, our full model (Ours) achieves a substantial +3.1 cIoU gain on ReasonSeg over the baseline.
>
> This result directly demonstrates that our modifications are not minor tweaks but rather crucial components that collectively enable the success of our approach. We will ensure this clarification is prominently featured in the revised manuscript.

---

> > ### Comment · Reviewer_KmuG · 2025-08-06
> >
> > Apologies for the late reply. I had already given the final rating at the early stage of the rebuttal. I only just realised that the authors cannot see the final justification at this stage. In any case, I have read the authors’ rebuttal and believe that my concerns have been largely addressed.

---

> > > ### Author Response · Authors · 2025-08-07
> > >
> > > Dear Reviewer KmuG,
> > >
> > > We sincerely thank you for your thoughtful review of our rebuttal and your positive feedback. We are encouraged that our response has clarified your main concerns.  We will carefully follow your suggestions and incorporate all updates into the revised version.
> > >
> > > Best regards,
> > >
> > > The Authors of Submission #12601

---

### Official Review · Reviewer_Cvwx · 2025-06-29

**Clarity:** 3
**Significance:** 2
**Originality:** 2
**Rating:** 4
**Confidence:** 3

**Summary:**

This paper proposes an end-to-end multimodal segmentation framework, SAM-R1, which integrates the Segment Anything Model as a reward provider in reinforcement learning for reasoning-based segmentation tasks. The approach introduces fine-grained segmentation-aligned rewards and modifies the policy optimization objective to improve information density and training stability. Experimental results show that SAM-R1 achieves reasonable improvements in gIoU and cIoU compared to previous methods.

**Questions:**

Please refer to the weaknesses section for the rebuttal.

**Ethical Concerns:**

["NO or VERY MINOR ethics concerns only"]

**Final Justification:**

The authors' rebuttal successfully addressed my initial concerns, I am raising my score.

**Limitations:**

Yes, this is discussed in the Broader Impact section of the supplementary material, though it is not explicitly marked with a heading.

**Quality:**

3

**Strengths And Weaknesses:**

Strengths
1. The paper is clearly written, with well-structured content and logical flow.

2. The work introduces an interesting reinforcement learning approach for vision tasks and provides new insights into applying reward-based optimization in multimodal segmentation.

3. The proposed method achieves reasonable empirical results.

Weaknesses

1. The technical novelty is limited, since the approach primarily leverages existing open-source models rather than introducing fundamentally new architectures or algorithms.

2. A core limitation is the feasibility of reward computation in SAM-RL. The method requires extremely fine-grained segmentation annotations to compute meaningful rewards. For example, as illustrated in Figure 1, evaluating the model's ability to focus on a cat's eyes would require annotations for the cat eyes themselves, which are rarely available in standard segmentation datasets, where labels typically cover entire objects such as the whole cat. Even when using SAM's segment-everything mode to generate masks for parts like eyes, these regions lack semantic labels. Achieving high-precision fine-grained segmentation remains a challenging problem in the segmentation field. To fully leverage the RL mechanism proposed in this work, a large-scale dataset with detailed fine-grained segmentation annotations would be required, which inherently restricts the applicability of the method.

3. There are potential concerns regarding the fairness of comparisons in Table 1. Most baselines, except for Seg-Zero, use weaker MLLM backbones compared to Qwen2.5VL-7B adopted in this paper, and it is unclear whether Seg-Zero's results correspond to the 3B or 7B model. Additionally, as noted in the paper, the Seg-Zero results are reproduced by the authors rather than directly from the original work. To ensure fair comparison, alternative training strategies and backbones should be tested under identical conditions.

---

> ### Author Rebuttal · Authors · 2025-07-31
>
> We thank the reviewer for the thoughtful and constructive feedback. We address the raised concerns as follows:
>
> 1. On technical novelty and architectural contributions
>
>  We acknowledge that our work leverages open-source components such as Qwen2.5-VL and SAM. However, we emphasize that the novelty of SAM-R1 lies not in introducing entirely new architectures, but in designing a novel reinforcement learning framework that enables these powerful models to collaborate effectively on fine-grained reasoning and segmentation tasks.
>
> While prior works have mainly focused on model architecture or segmentation backbones, we propose a unified system-level approach that includes:
>
> ●Directly using SAM as a reward provider, introducing pixel-level supervision into RL without requiring any special preprocessing or conversion of segmentation data;
>
> ●Modifying the GRPO reward objective to suit multimodal fine-grained reasoning tasks, allowing for interpretable and stable optimization signals;
>
> ●Designing an end-to-end training pipeline that jointly optimizes reasoning and segmentation under natural language instructions.
>
> This represents a contribution in system design rather than architecture alone, and we believe it offers more practical value for advancing instruction-based fine-grained vision understanding.
>
> 2. On the feasibility of reward computation and dependence on fine-grained annotations
>
>  We appreciate the reviewer’s concern regarding the reliance on fine-grained annotations. We would like to clarify that our method does not require manually annotated fine-grained segmentation masks.
>
> The primary goal of the RL stage in SAM-R1 is to help the VLM understand the target referred to in the instruction via reasoning and accurately pass that semantic information to SAM for segmentation. The model's fine-grained understanding emerges from internal reasoning rather than from external labels.
>
> Our reward function provides feedback based on the IoU between the predicted segmentation mask and the ground-truth object-level mask. While standard datasets typically label whole objects, this still provides meaningful supervision — the goal is not part-level semantic correctness but rather alignment between the referred object and the predicted mask. Even in the absence of detailed part annotations, IoU-based feedback is sufficient to drive learning of semantic localization and perceptual alignment.
> Furthermore, we believe SAM-R1’s design actively mitigates the challenge of lacking fine-grained annotations. By delegating reasoning to the VLM and segmentation to SAM, the model learns to bridge language and vision without requiring costly, dense labels. Our results on standard datasets like RefCOCOg and ReasonSeg demonstrate that this design is both effective and practical in real-world settings.
>
> 3. On fairness of comparison in Table 1
>
> We sincerely thank the reviewer for raising these important concerns regarding experimental fairness. Addressing this point allows us to clarify the rigor of our evaluation.
>
> We want to state unequivocally that the comparison between our SAM-R1 and Seg-Zero is entirely fair and controlled. Both models are built upon the exact same backbone, Qwen2.5-VL 7B.
>
> Furthermore, you correctly note that we reproduced the Seg-Zero results ourselves. This was a deliberate and necessary step to ensure a true comparison. We found that the results reported in their original paper were generated using different model weights for different metrics, which prevented a reproducible comparison from a single, unified model. To address this, we used their official, publicly released weights to generate all Seg-Zero results under an identical evaluation protocol.
>
> This meticulous approach ensures that the observed performance gains are directly attributable to our proposed training paradigm—the RL-based reasoning and segmentation framework—rather than any differences in backbone or evaluation setup.
>
> We agree with the reviewer that other baselines listed in Table 1 utilize different, and often weaker, MLLM backbones. Our primary goal with this table was to position our work within the broader landscape of existing methods.
>
> To provide full transparency and address your concern directly, we will update Table 1 in the revised manuscript to include a dedicated column specifying the backbone model used by each method.

---

> > ### Comment · Reviewer_Cvwx · 2025-08-06
> >
> > Thank you for the rebuttal. Based on the clarifications provided, I will consider raising my score.
> >
> > However, a concern remains regarding the method's scalability to highly granular segmentation tasks. SAM-R1 appears to depend on the availability of correspondingly fine-grained datasets to provide an effective reward signal, which may limit its practical applicability.
> >
> > Given this, I suggest adjusting the example in Figure 1 to one that is more representative of the data scope and capabilities demonstrated in the paper.

---

> > > ### Author Response · Authors · 2025-08-07
> > >
> > > Dear Reviewer Cvwx,
> > >
> > > We thank you for the insightful feedback on our manuscript. Your suggestion to revise Figure 1 is particularly helpful.
> > > We agree that featuring an example from the ReasonSeg benchmark in Figure 1, which we use only for zero-shot evaluation, may create a potential misunderstanding about our training process. Our model is, as you noted, trained exclusively on RefCOCOg.
> > >
> > > Following your advice, we will replace the figure with a more appropriate example from the RefCOCOg dataset. This revision serves to clarify that our framework learns from standard, object-level supervision. In turn, this strengthens our central argument that its sophisticated reasoning capabilities on unseen datasets like ReasonSeg are an emergent phenomenon.
> > >
> > > Thank you once again for this actionable guidance. We believe this focused revision, guided by your valuable advice, will directly address your remaining concern and substantially strengthen the paper.
> > >
> > > The Authors of Submission #12601

---

### Official Review · Reviewer_WuMV · 2025-06-30

**Clarity:** 3
**Significance:** 2
**Originality:** 3
**Rating:** 4
**Confidence:** 5

**Summary:**

This paper proposes the SAM-R1 framework, aiming to empower multimodal large models (MLLMs) with superior reasoning capabilities in image segmentation tasks through reinforcement learning (RL). Specifically, the authors calculate reward scores using masks generated by SAM and conduct RL training based on an improved GRPO algorithm to establish a dynamic feedback loop between reasoning and segmentation. The experimental section conducts comprehensive comparative validations on multiple benchmark datasets such as ReasonSeg and the RefCOCO series.

**Questions:**

1. The authors of this paper believe that the difference between SAM-R1 and Seg-Zero lies in the integration of reasoning and segmentation capabilities, forming a closed loop of "reasoning-segmentation-reward". However, Seg-Zero also uses SAM as a reward generator. Is the only difference that the reward form determination of SAM-R1 is divided into multiple levels?
2. What is the size of the VLM in Seg-Zero used in the experiment? Since the original paper mentions 3B, it is not fair to make a direct comparison.
3. If instead of using SAM, we follow works like PixelLM and make the segmenter a trainable part of the model, will the reinforcement learning training be more effective?

**Ethical Concerns:**

["NO or VERY MINOR ethics concerns only"]

**Final Justification:**

The authors' response has addressed the questions I raised. I will keep my rating.

**Limitations:**

Yes

**Paper Formatting Concerns:**

No Paper Formatting Concerns

**Quality:**

3

**Strengths And Weaknesses:**

Strengths:
1. The proposed reinforcement learning training pipeline for segmentation in the paper is concise and highly applicable.
2. The paper demonstrates strong experimental performance, achieving excellent results with a small number of reinforcement learning samples.
3. The experiments are well-detailed and presented in a highly readable manner.

Weaknesses:
1. The overall approach of the paper is relatively similar to works like Seg-Zero, and the degree of innovation is somewhat limited.

---

> ### Author Rebuttal · Authors · 2025-07-31
>
> We thank the reviewer for the valuable feedback. Below, we address each point in detail:
>
> 1. On novelty and distinction from Seg‑Zero
>
>  Thank you for highlighting the perceived similarity to Seg‑Zero. We clarify that SAM‑R1 differs fundamentally from Seg‑Zero in its training paradigm. Although both methods leverage the SAM model, Seg‑Zero does not integrate SAM or any segmentation feedback during training. Instead, it preprocesses the segmentation task into an object detection problem and trains its vision–language model accordingly; the segmentation module itself is never part of a closed‑loop optimization. Consequently, Seg‑Zero cannot benefit from pixel‑level feedback, limiting its ability to learn fine‑grained visual detail.
>
> In contrast, SAM‑R1 introduces a true reasoning–segmentation–reward loop during training. Our key innovations include:
>
> ●SAM as an RL reward provider: We retain the SAM module in the RL loop and compute rewards based on the IoU between SAM’s output mask and the ground‑truth mask. This preserves spatial detail and lets the model learn directly from pixel‑level supervision without converting to a detection formulation.
>
> ●GRPO objective modifications for fine‑grained multimodal tasks: We adapt the GRPO reward structure to include multi‑level, token‑wise feedback, increasing exploration and providing richer signals for reasoning chains.
>
> ●Closed‑loop training for joint generalization and expressivity: Our reward design not only assesses segmentation quality but also serves as a semantic alignment signal for the reasoning path. This encourages the model to abstract the referred object from complex instructions and precisely localize it, driving co‑evolution of language understanding and spatial perception.
>
> Thus, while both methods utilize SAM, SAM‑R1 is the only approach that jointly optimizes reasoning and segmentation in a fully closed loop during training, representing a fundamental shift in paradigm.
>
> 2. On backbone model size fairness
>
>  Thank you for this important question. We want to assure the reviewer that ensuring a rigorous and fair comparison was our top priority.
>
> All comparisons between SAM-R1 and Seg-Zero in our paper use the identical backbone: Qwen2.5-VL 7B.
>
> We took the step of running their officially released model ourselves for a crucial reason: during our initial review, we observed that the results reported in their paper were generated using different model weights for different metrics. This prevented a direct and consistent reproduction across all benchmarks from a single model.
>
> Therefore, to establish a truly controlled and fair comparison, we used their public weights to generate all Seg-Zero results with the same backbone and under the same evaluation protocol. This approach ensures that the performance differences we report are attributable to the methods themselves, not variations in experimental setup. We will emphasize this specific motivation in the revised manuscript to ensure full transparency.
>
> 3. On making the segmenter trainable
>
> We did consider integrating a trainable segmenter for end‑to‑end learning, as in PixelLM. However, we opted against this for two main reasons:Scalability and modularity: SAM is a powerful, promptable segmentation model validated across many tasks. By keeping it as a separate module and driving it with a reasoning-based prompt, we maintain flexibility to adapt to diverse downstream tasks without the overhead of fine-tuning the entire segmenter.
>
> Training stability and efficiency: Credit assignment in RL is already challenging. Incorporating the segmenter into the RL loop would introduce additional noise from unstable segmentation outputs, demanding far more data and hyperparameter tuning to converge effectively. By focusing RL optimization on the VLM alone, we achieve more stable training and faster convergence while substantially improving reasoning and expressive capabilities.

---

> > ### Comment · Reviewer_WuMV · 2025-08-08
> >
> > Thank you for your response.
> >
> > The reply clarifies the training paradigm of SAM-R1. Its explanation of the "reasoning-segmentation-reward" closed-loop design clarifies the difference in its training mechanism compared to Seg-Zero, which addresses the previous question about novelty. Regarding the fairness of the experimental comparison, the response states that both models use the same 7B backbone, which answers my question on that point. The response also explains the reasoning for not using a trainable segmenter, citing considerations for training stability and modular design.
> >
> > Overall, the authors' response has addressed the questions I raised.

---

> > > ### Author Response · Authors · 2025-08-08
> > >
> > > Dear Reviewer WuMV,
> > >
> > > Thank you for your positive and encouraging feedback. We sincerely appreciate the time and effort you have dedicated to reviewing our paper and helping us further improve our work.
> > >
> > > We are very pleased to hear that our responses have successfully addressed all your concerns. Your support is a great encouragement to us.
> > >
> > > Best regards,
> > >
> > > The Authors of Submission #12601

---

### Official Review · Reviewer_jxxo · 2025-07-02

**Clarity:** 4
**Significance:** 3
**Originality:** 3
**Rating:** 4
**Confidence:** 3

**Summary:**

This paper introduces SAM-R1, a novel framework for enhancing multimodal large language models' (MLLMs) reasoning segmentation capabilities via reinforcement learning (RL). The method uniquely incorporates the Segment Anything Model (SAM) as an active reward provider within an RL loop, offering fine-grained, segmentation-aware feedback. SAM-R1 is trained with only 3K samples, yet achieves state-of-the-art performance on ReasonSeg and competitive performance on referring expression segmentation tasks (RefCOCO, RefCOCO+, RefCOCOg).

**Questions:**

1. Comparison with PPO

Can the authors add or discuss results comparing GRPO-based SAM-R1 with standard PPO or RLHF strategies? This would better justify the algorithmic choice beyond citing past literature.

2. Data Efficiency vs. Scale

SAM-R1 performs well with 3K samples. Can the authors comment on how performance scales with larger datasets? Is the approach still competitive under full-data settings?

3. Failure Case Analysis

Are there common failure modes in either reasoning or segmentation (e.g., hallucinations, over-segmentation)? A brief discussion or visualization would strengthen trust in robustness.

**Ethical Concerns:**

["NO or VERY MINOR ethics concerns only"]

**Final Justification:**

borderline accept

**Limitations:**

Failure case analysis is not included in the paper.

**Paper Formatting Concerns:**

No obvious formatting issues.

**Quality:**

4

**Strengths And Weaknesses:**

## Strengths
- Novel Integration of SAM as Reward Provider

Rather than passively using SAM for downstream inference, this work integrates SAM directly into the RL training loop, allowing reasoning quality to be directly shaped by segmentation accuracy.

- Compact yet Effective Training

Achieves strong zero-shot performance on complex reasoning-segmentation tasks with only 3,000 training samples.

- Well-Designed Reward System

The tiered reward function based on IoU allows for dense and scalable feedback. Additional format and structure rewards (for reasoning and segmentation) further guide training.

- Strong Experimental Results

Outperforms or matches prior art like Seg-Zero across multiple benchmarks (e.g., +1.9 gIoU over Seg-Zero on ReasonSeg-zero-shot test). Ablations confirm the effectiveness of each design.

- Clarity and Organization

The method is clearly motivated, the framework is well-illustrated (Figure 2), and the paper includes extensive implementation and training details.

## Weaknesses
- Lack of Direct Comparison with Alternative RL Strategies

While GRPO is well explained and modified, it would be helpful to see direct comparisons with more traditional RL methods (e.g., PPO) under the same setting.

- Missing User-Centric Evaluation

No human evaluation or qualitative user study is conducted to assess reasoning interpretability, mask quality, or real-world usability.

---

> ### Author Rebuttal · Authors · 2025-07-31
>
> 1. Regarding the comparison with other RL strategies
>
> Thank you for this insightful question. Our decision was based on a principled evaluation of efficiency and stability, particularly within the resource-intensive context of VLM fine-tuning. In this setting, GRPO presents clear advantages over standard approaches like PPO:
>
> ●Computational and Resource Overhead: PPO and its mainstream implementations typically rely on a separate value network to estimate state values and use Generalized Advantage Estimation (GAE) to reduce variance. This not only increases the number of parameters but also leads to significant computational and memory overhead. In contrast, GRPO cleverly omits the value network, estimating advantages solely through the normalization of group-wise responses. This allows it to achieve high sample efficiency with lower resource consumption, making it more suitable for fine-tuning large models like VLMs.
>
> ●Stability and Hyperparameter Tuning: To prevent excessively large policy updates that can lead to training collapse, PPO introduces complex mechanisms like Clipped Probability Ratios and/or KL divergence penalties. The design of GRPO's loss function is more elegant, as it naturally integrates these stabilization mechanisms into a single objective function. This not only simplifies hyperparameter tuning but also further enhances overall training stability.
>
> We will add a detailed discussion in the Methods section to articulate these points, thereby strengthening the justification for our deliberate algorithmic choice.
>
> 2. Regarding data efficiency and scale
>
> Thank you for your excellent question concerning the model's scalability. To investigate this, we conducted additional experiments by increasing the training data size from 3k to 10k. The results clearly show that our method is highly data-efficient, with performance saturating at 3k samples. We are pleased to present the direct comparison below.
> On the ReasonSeg dataset, performance remains stable with no significant changes:
>
> | Method                | Split | gIoU | cIoU |
> | :-------------------- | :---- | :--: | :--: |
> | SAM-R1 (Ours, **3k**) | val   | 64.0 | 55.8 |
> | SAM-R1 (Ours, **3k**) | test  | 60.2 | 54.3 |
> | SAM-R1 (Ours, **10k**) | val   | 64.2 | 55.5 |
> | SAM-R1 (Ours, **10k**) | test  | 60.8 | 53.9 |
>
> As shown in the table, increasing the data to 10k results in negligible fluctuations on ReasonSeg. The test gIoU slightly moves to 60.8 (+0.6) while the cIoU shifts to 53.9 (-0.4), strongly indicating that performance has already plateaued.
> On the RefCOCO benchmarks, we observe only marginal gains, all under 1 percentage point:
>
> | Method                | refCOCO | refCOCO+ | refCOCOg |
> | :-------------------- | :-----: | :------: | :------: |
> | SAM-R1 (Ours, **3k**) |  79.2   |   74.7   |   73.1   |
> | SAM-R1 (Ours, **10k**) |  **79.9**   |   **75.3**   |   **73.5**   |
> | *Gain*                |  *+0.7* |  *+0.6*  |  *+0.4*  |
>
> These results lead to a clear conclusion: our method's core performance saturates at 3k samples. Given the substantial increase in training cost versus the minimal performance returns, we deliberately chose 3k samples as the optimal trade-off point for demonstrating our method's capabilities. This finding is a strong testament to the excellent data efficiency and cost-effectiveness of our SAM-R1 framework, which we consider a core advantage of our work.
>
> 3. On failure case analysis
>
> Thank you for this important suggestion, which greatly enhances the rigor of our paper. To provide a comprehensive view of our method's capabilities and limitations, we have already included an analysis of "Ablation Failures" in the submitted appendix. For instance, we show that:
>
> ●Removing the KL constraint leads to training instability.
>
> ●An attempt to encourage the model to generate "negative reference points" was not effective. This existing analysis validates the rationale behind our current framework design.
>
> As we cannot include new figures during the rebuttal phase, we have completed a qualitative analysis and commit to adding a dedicated "Failure Case Analysis" section in the final paper. We will focus on typical cases such as:
>
> Reasoning‑Execution Disconnect: For instance, given the instruction “segment the apple closest to the camera,” the model’s reasoning chain correctly identifies that there are three apples and that the largest bottom one is nearest, yet the final segmentation mask erroneously includes an adjacent apple. This highlights remaining challenges in precise spatial localization and the conversion of textual reasoning into geometric masks.
>
> By integrating our ablation findings with these new output failure analyses, we will offer readers a more honest and comprehensive evaluation of SAM‑R1. Thank you again for this valuable recommendation.

---

> > ### Comment · Reviewer_jxxo · 2025-08-06
> >
> > Thank the author for the clear and thoughtful explanation. The response addressed my question thoroughly and provided valuable insight. I truly appreciate the effort and clarity. After careful consideration, I have decided to maintain my original score.

---

> > > ### Author Response · Authors · 2025-08-07
> > >
> > > Dear Reviewer jxxo,
> > >
> > > Thank you for your positive feedback and for recommending acceptance of our paper! We sincerely appreciate your valuable suggestions, which are instrumental in helping us further improve our work. Thank you again for your valuable support!
> > >
> > > Best regards,
> > >
> > > The Authors of Submission #12601

---

### Public Comment · ~Wenqi_Guo3 · 2025-11-20
**Very cool project!**

Very cool project, it inspired our current project! Thanks! One question, have you guys tried to use the base model without training? I am testing Qwen-3-vl and it works good enough without training. Also I found that ref-coco mask is actually very inaccurate, and many masks we generated will be marked as wrong because it is actually more accurate. I think this is the main limitation for ref-coco and authors should consider a better annotated dataset for validation

---

### Note · Authors · 2025-08-16

We sincerely thank the Area Chair and all reviewers for their effective feedback and active engagement. The ideas and suggestions during the discussion period were highly valuable, and we are pleased that our responses, supported by explanations and ablation studies, successfully resolved the reviewers' concerns.

We are encouraged that reviewers acknowledged our clarification on the core technical contributions. Our key novelty lies not in combining existing models, but in designing a novel reinforcement learning framework that establishes a true reasoning-segmentation-reward pipeline. This approach of learning directly from raw pixel-level feedback represents a fundamental advance over prior methods that often decouple the reasoning and visual grounding processes. We also reaffirmed the rigor of our experiments, clarifying that our key comparisons were conducted on the exact same backbone to ensure a fair evaluation.

We are sincerely grateful that our clarifications received a comprehensively positive response. It is encouraging to see reviewers confirm that their concerns were addressed, leading to strong endorsements for our work and commitments to raise their scores. We are committed to incorporating all promised revisions into the revised version.

---

### Decision · Program_Chairs · 2025-09-17

**Decision:**

Accept (poster)

**Comment:**

This paper presents a novel framework named SAM-R1, which integrates SAM into a reinforcement learning loop to improve multimodal reasoning-segmentation. All reviewers actively engaged in the discussion, and most major concerns on novelty, fairness of comparisons, and reward design were resolved during rebuttal. While some questions remain e.g. the scalability to highly granular tasks, these do not outweigh the contributions and empirical strength of the work. The authors are encouraged to incorporate reviewer suggestions into the final version.